# Using experimental results of protein design to guide biomolecular energy-function development

Hugh K. Haddox[1,2,*], Gabriel J. Rocklin[1,2,3,4], Francis C. Motta[5], Devin Strickland[6], Samer F. Halabiya[6], Cameron Cordray[6], Hahnbeom Park[1,2], Eric Klavins[6], David Baker[1,2,7], Frank DiMaio[1,2,*]

**1** Department of Biochemistry, University of Washington, Seattle, Washington, United States of America, **2** Institute for Protein Design, University of Washington, Seattle, Washington, United States of America, **3** Department of Pharmacology, Northwestern University Feinberg School of Medicine, Chicago, Illinois, United States of America, **4** Center for Synthetic Biology, Northwestern University, Evanston, Illinois, United States of America, **5** Department of Mathematics and Statistics, Florida Atlantic University, Boca Raton, Florida, United States of America, **6** Department of Electrical and Computer Engineering, University of Washington, Seattle, Washington, United States of America, **7** Howard Hughes Medical Institute, University of Washington, Seattle, Washington, United States of America

* hhaddox@fredhutch.org (HKH); dimaio@uw.edu (FD)

## Abstract

Computational models of macromolecules have many applications in biochemistry, but physical inaccuracies limit their utility. One class of models uses energy functions rooted in classical mechanics. The standard datasets used to train these models are limited in diversity, pointing to a need for new training data. Here, we sought to explore a new paradigm for training an energy function, where the Rosetta energy function was used to design *de novo* proteins. Experimental results on these designs were then used to identify failure modes of design, which were subsequently used as a "guiding principle" to retrain the energy function. Specifically, we examined a diverse set of *de novo* protein designs experimentally tested for their ability to stably fold, identifying unstable designs that were predicted to be stable by the Rosetta energy function. Using deep mutational scanning, we identified single amino-acid mutations that rescued the stability of these designs, providing insight into common failure modes of the energy function. We identified one key failure mode, involving steric clashing in protein cores. We identified similar overpacking when using Rosetta to refine high-resolution protein crystal structures, quantified the degree of overpacking, and refit a small set of energy-function parameters to better recapitulate native-like packing. Following fitting, we largely eliminated the failure mode in the refinement task, while retaining performance on other benchmarks, resulting in an updated version of the Rosetta energy function. This work shows how learning from protein designs can guide energy-function development.

**Data availability statement:** See https://github.com/Haddox/design_guided_optE for input data, output data, and code. The Rosetta source code is freely available for academic use in the Rosetta software suite at https://rosetta-commons.org/. We have added the beta_jan25 energy function to the Rosetta source code. The energy function can be invoked using the "-beta_jan25" flag (see https://docs.rosetta-commons.org/docs/latest/full-options-list). To implement the precise energy function used in the dualoptE protocol, also include the flag "-score::count_pair_hybrid false".

**Funding:** This work was supported by DARPA Synergistic Discovery and Design (FA8750-17-C-0219 to HKH, GJR, DS, SFH, CC, HP, EK, and DB; FA8750-17-C-0054 to FCM subcontracted through Duke University) and by the Washington Research Foundation (HKH). The funders had no role in study design, data collection and analysis, decision to publish, or preparation of the manuscript.

## Author summary

Computational models of macromolecules have many applications, such as predicting structures, predicting mutational effects, or designing new proteins. One type of model uses an energy function to explicitly model the physical forces at play. These models are trained using experimental data. However, available training data are limited in number and diversity, prompting a need for new sources of training data. In this paper, we explore a new paradigm for training an energy function, which involves using the energy function to design *de novo* proteins and then learning from which designs succeed and which fail when experimentally tested in the lab. We used experimental data to learn common failure modes in design, identified examples of a failure mode in a high-quality benchmark, and then used this benchmark to retrain the energy function. Using this strategy, we identified and largely resolved a bias in the Rosetta energy function that involved energetically unfavorable steric clashes in protein cores. Overall, this work helps establish a framework for how learning from design can be used to guide the development of macromolecular models.

## Introduction

Computational macromolecular models have many applications, such as predicting and refining structures [1–5], designing new proteins [5–8], and predicting mutational effects [9–11]. Models using deep learning have made tremendous advances in structure prediction and design [1–3,6–8]. However, despite these advances, there are still some areas where there is need for improved models, including predicting mutational effects and predicting biomolecular interfaces–particularly host/pathogen and antibody/antigen interfaces where sequence coevolution cannot be used to guide prediction. Further, such models are limited in their ability to model nonstandard chemistries or amino-acid modifications for which there is limited training data. Classical models may generalize better in these tasks, as well as providing insight into the underlying chemical interactions giving rise to function. Although such models have been refined over many years, they still have physical inaccuracies, owing to the limited datasets used to train them. In the past, three main sources of data have been used in training: small-molecule thermodynamic data, high-resolution protein crystal structures, and effects of mutations on protein-folding free energies [12–18]. One study showed that supplementing protein-based training data with small-molecule data greatly improved an energy function's accuracy on a variety of biophysical benchmarks [18]. This result suggests that a promising strategy for continued energy-function development is to increase the diversity of training data. Yet, energy functions are typically trained on nearly all available data from the above sources. And it is nontrivial to substantially increase the diversity and size of these datasets.

To address this limitation, we sought to explore a new paradigm for energy-function optimization: learning from *de novo* protein design. Using an energy function

to design new proteins gives a large amount of freedom to explore sequences and structures, which is useful because it can expose flaws that might otherwise be difficult to detect. Identifying common failure modes in design could help diagnose underlying inaccuracies in an energy function. In turn, retraining an energy function using experimental data on designs could allow it to learn from its past successes and failures. Multiple studies have used high-throughput experimental testing of designs to uncover specific biophysical properties that correlate with design success [19–22]. One study used such data to retrain an energy function in a highly specific design task [20]. However, it did not test whether this energy function generalized to other design and molecular-modeling tasks. Here, we sought to use experimental data on a diverse set of *de novo* protein designs to guide the general development of the Rosetta energy function.

## Results

### Sources of experimental data on designs

The Rosetta energy function has been used to design a wide variety of *de novo* proteins, many of which have then been experimentally characterized to identify successful and failed designs. We sought to use such data to guide further energy function development. We focused on three main datasets. The first dataset is from Rocklin et al. [19], which used Rosetta to computationally design thousands of miniproteins (soluble proteins, ~40 amino acids in length), and then experimentally estimated the stability of each design using a high-throughput assay that quantifies the sensitivity of miniproteins to proteolysis. Designs spanned a wide range of stabilities with hundreds of both successful (stable) and failed (unstable) designs. The second dataset is newly published as part of this study (described below) and uses the same high-throughput assay to identify single amino-acid mutations that stabilize failed designs from Rocklin et al. A third dataset involves 17 crystal structures of Rosetta designs, which we compiled across several studies and which span several protein folds and design methodologies (S1 Table) [23–33]. These crystal structures often had Angstrom-level differences from their corresponding computational design models, and suggest potential failure modes in designing precise structures. We hypothesized that Rosetta's energy function could learn from past successes and failures by using these data to retrain it, but we lacked a clear methodology for doing so. The next section describes our efforts to establish one.

### Single amino-acid mutations that rescue failed protein designs suggest failure modes

While our initial inclination was to directly encode these data in our energy-function optimization framework [18] (Fig 1, top track), initial experiments showed difficulty in getting this fitting to generalize to unseen data. Therefore, we pivoted to

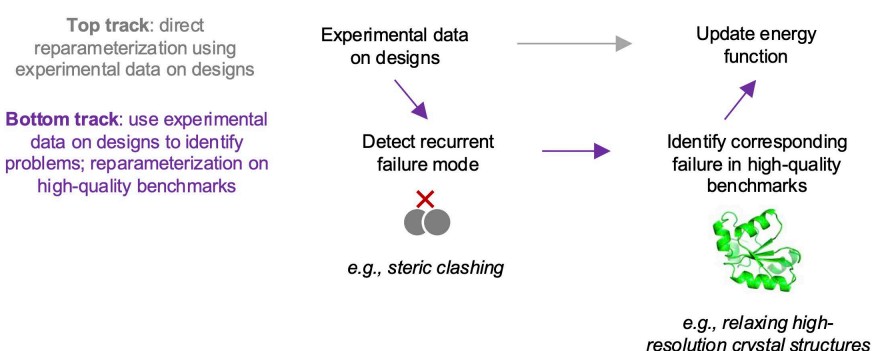

**Fig 1. Overview of approaches for learning from experimental testing of designs.** The top and bottom tracks show our two basic approaches. Following the bottom track, we used experimental data on Rosetta designs to identify a recurrent failure mode involving steric clashing. We identified examples of this failure mode in a high-quality benchmark involving refining high-resolution crystal structures. We used this benchmark and others to update the Rosetta energy function.

using an *indirect* learning approach (Fig 1, bottom track). This approach involved: i) using the experimental data to identify a common failure mode in design, ii) identifying examples of this failure mode in a high-quality benchmark, and iii) using this benchmark and others to reparameterize the energy function.

To start, we focused on the high-throughput miniprotein stability dataset from Rocklin et al. Rosetta's energy function has a limited ability to predict the stability of these designs (Figs 2A and S1A). Rocklin et al. used machine learning to identify biophysical features that correlate with design success and suggest certain failure modes. To search for additional failure modes, we identified a set of outlier designs from Rocklin et al. that are unstable according to the high-throughput stability assay, but were predicted to be stable by machine-learning models trained to predict stability scores from on a wide array of biophysical features, including Rosetta energies (S1A Fig). The outlier designs tended to have much more favorable Rosetta energies than other unstable designs (see red dots in Figs 2A; S1A). We hypothesized that each of these designs are adjacent in sequence space to a highly stable sequence.

To test this hypothesis, we used deep mutational scanning (DMS) to experimentally measure the effects of all single amino-acid mutations on the stability of 21 outlier designs spanning three miniprotein topologies. We carried out the DMS experiment using the high-throughput stability assay from Rocklin et al. In brief, this experiment involves expressing mini-protein DMS libraries on the surface of yeast cells, incubating cells with protease to digest unstable miniproteins, enriching for cells with intact miniproteins using fluorescence-activated cell sorting, and deeply sequencing libraries before and after selection to quantify the enrichment or depletion of each variant. We independently performed the selection step with two different proteases and six increasing concentrations of protease, allowing us to estimate an $EC_{50}$ for each variant to each protease. We also performed biological replicates of the entire experiment for 19 of the 21 outlier designs (the 2 other outlier designs were tested separately with a single replicate experiment; see Methods).

As hypothesized, a subset of mutant designs were highly enriched upon selection, indicating increased stability. For instance, Fig 2B shows enrichment values of all amino-acid variants at a specific site in one of the outlier designs. The original design tended to become more and more depleted as the protease concentration used in selection increased (blue line), and nearly all amino-acid variants at this site followed this trend (gray lines). However, two mutants were strongly enriched (orange and red lines).

To quantify the effects of mutations on folding stability, we computed a "stability score" for each variant as the log ratio of the variant's observed $EC_{50}$ vs. the predicted $EC_{50}$ of the variant in an unfolded state [19]. Positive stability scores indicate the observed $EC_{50}$ is higher than the predicted one, suggesting some level of folding stability. Nearly all outlier designs had low stability scores near zero in our data (S1D Fig), consistent with the original measurements from Rocklin et al. And most single amino-acid mutations to these designs had neutral effects on the design's stability scores (Fig 2C). However, a subset of mutations had large positive or negative effects, with the positive (stabilizing) tail of the distribution extending further than the negative one (Figs 2C; S1E). Mutations to outlier designs with intermediate initial stability scores in our data followed a similar pattern, but had a larger tail in the negative direction due to their greater initial stability (S1F Fig).

Across all outlier designs, ~1% of mutations (156 out of ~16,000 mutations tested) were highly stabilizing (Δ stability score > 1.0). We term these "rescue mutations". Most designs had at least one rescue mutation, while some had several (Fig 2D). Although some of these large effects may be due to noise, many rescue mutations involved similar biophysical changes, suggesting a real effect.

The most common pattern of rescue mutations involved changing a polar amino acid to a nonpolar one (Fig 2E). Fig 2G shows examples of this pattern repeated across five sites from three designs. These mutations often occurred at the boundary of a design's hydrophobic core (Fig 2H), where mutations to nonpolar amino acids would lead to increased levels of buried nonpolar surface area – a feature that Rocklin et al. identified to be correlated with miniprotein stability [19]. Within a single design, such rescue mutations are often confined to one or a few polar sites (S2 Fig), arguing against an indiscriminate mechanism of stabilization, such as increased propensity for nonspecific aggregation. This class of rescue

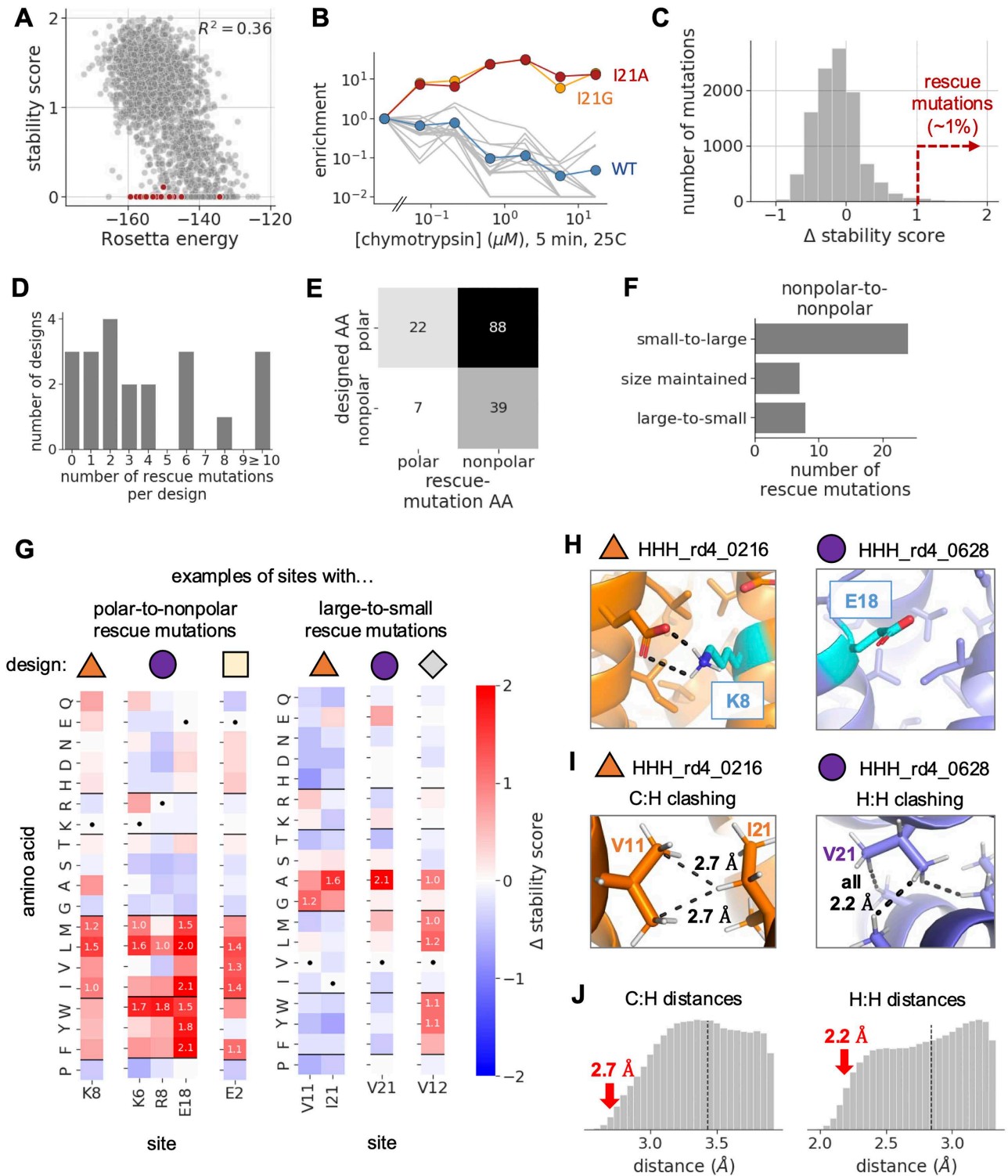

**Fig 2. Deep mutational scanning identifies single amino-acid mutations that rescue unstable designs. A)** For each three-helix bundle miniprotein design from Rocklin et al., this plot shows the design's Rosetta energy (x-axis) compared to its experimentally measured stability score (y axis; negative values are clipped at zero). Red dots show outlier designs chosen for deep mutational scanning. S1A Fig shows similar plots for all three miniprotein topologies we analyzed. **B)** Enrichment values for all amino-acid variants at site 21 of the HHH_rd4_0216 design, from a selection performed with

chymotrypsin. Enrichment values are computed as a variant's frequency in the library after selection divided by its frequency in the initial unselected library. Each line shows data for a different variant. **C)** The distribution of mutational effects on the stability score of the 14 outlier designs with an initial stability score of <0.5, as measured in this study. Positive effects indicate that the mutation increased the design's stability, while negative effects indicate the opposite. About 1% of mutations had effects >1.0. **D)** The number of rescue mutations per design. **E)** Classification of rescue mutations based on the polar or nonpolar identity of designed vs. mutant amino acids. **F)** Among rescue mutations that involve replacing one nonpolar amino acid with another, this plot classifies mutations based on the size of the designed amino acid compared to the size of the mutant one, reporting the number of mutations in each category. **G)** Heatmaps show mutational effects, with each column showing effects at a specific site from a specific design. Sites are grouped by design (colored shapes) and biophysical patterns of rescue mutations. Boxes corresponding to rescue mutations have numbers reporting the mutation's effect. Dots show boxes corresponding to originally designed amino acids. **H)** The structural context of two example sites with polar-to-nonpolar rescue mutations in the indicated designs (shape labels match panel **G**). **I)** The structural context of three example sites with large-to-small rescue mutations in the indicated designs. The left example shows clashing between carbon atoms from V11 with a hydrogen atom from I21, with dashed lines measuring interatomic distances. The right shows clashing between hydrogen atoms from V21 and three neighboring sidechains, with dashed lines measuring interatomic distances, all of which are 2.2 Å. The structures shown were relaxed in Cartesian space with the Rosetta energy function. **J)** Smoothed distributions of interatomic distances between pairs of carbon and hydrogen atoms (left) or hydrogen atoms (right) from hydrophobic sidechains in 78 high-resolution crystal structures of native proteins, with hydrogen atoms placed by Rosetta. Vertical dashed lines show the sum of the van der Waals radii of the two atoms. Atoms closer than this line are "clashing". Distances are truncated at 0.5 Å above this line. Red arrows point to distances observed in miniprotein designs from panel **I**.

mutations suggests that the design protocol underweighted the hydrophobic effect, either due to Rosetta's energy function or the layer-based sampling procedure used to limit hydrophobic surface content.

Another class of rescue mutations involved mutations from one nonpolar amino acid to another (Fig 2E). Most mutations in this category involved mutating a smaller sidechain to a larger one (Fig 2F). Often, these mutations occurred at sidechains packed in the protein core, suggesting that large rearrangements would be needed to accommodate them. As above, it's possible these mutations are stabilizing because they increase the size of the hydrophobic core.

In comparison, a trend that was less common but more surprising involved nonpolar-to-nonpolar mutations that *decreased* sidechain size (Fig 2F). Fig 2G shows multiple examples of mutations in this category, which all involve mutating either a valine or isoleucine in the protein core to either an alanine or glycine. Contrary to the above pattern, these mutations would decrease the size of the core. Closer inspection of these mutations revealed that the designed sidechains are predicted to sterically clash with neighboring sidechains in the Rosetta structural models. Fig 2I shows example clashes in two designs. The first design has two sites with large-to-small mutations: V11 and I21. In the structure, carbon atoms from V11 clash with a hydrogen from I21. The second design has one site with large-to-small mutations, and hydrogen atoms from this site clash with hydrogens from three neighboring sidechains. These clashes are more severe than nearly all clashes observed in high-resolution crystal structures of native proteins (Fig 2J), suggesting that large-to-small mutations might be stabilizing because they relieve energetically unfavorable clashes in the computationally designed structure.

The large-to-small rescue mutations suggest that Rosetta's energy function might underweight the energetic cost of steric clashing, leading it to tolerate sidechains that clash with their surroundings. If Rosetta underweights the cost of steric clashing, one might expect clashes to be present in other miniprotein designs from Rocklin et al., especially unstable ones. Indeed, the miniprotein designs generally had elevated levels of clashing between atoms from nonpolar sidechains, as compared to levels in high-resolution crystal structures of native proteins (S3 Fig). To our surprise, both stable and unstable miniprotein designs had similarly elevated levels of clashing, suggesting it is a general feature of the designs. We focus on this putative failure mode for the remainder of the paper.

### Clashes in design models are often absent in corresponding crystal structures

The above results raise the question of whether designed clashes actually occur in reality. To investigate this question, we examined the set of Rosetta designs with experimentally resolved crystal structures described above (S1 Table). For each design model and corresponding crystal structure, we quantified interatomic distances between pairs of carbons

from nonpolar side chains. We then counted the number of atom pairs that were clashing (defined as their distance being shorter than the sum of their radii). This number was always lower in the crystal structure compared to the design model, with crystal structures having ~30% fewer clashing atom pairs on average (Fig 3A). The crystal structures also showed reduced levels of clashing between other atom pairs, including carbons from nonpolar sidechains and backbone oxygens (Fig 3B). In examining the structural changes that account for these trends, we found that many clashes were relieved through minor backbone rearrangements. However, some were relieved through much larger rearrangements that substantially altered the design's shape (Figs 3C; S4A). Thus, clashes in designs may be one factor that causes crystal structures to differ from corresponding design models, interfering with accurate protein design. In all, this section and the previous one suggest that Rosetta might be overly tolerant of steric clashing.

### Rosetta steric-clashing bias is observable in a high-quality benchmark

Following the workflow from Fig 1, we next sought to identify examples of the steric-clashing problem in a benchmark with high-quality structural data. We focused on a benchmark that involves refining high-resolution crystal structures using the Rosetta *relax* protocol. This benchmark computes distributions of distances between pairs of atom types before and after relaxing structures with Rosetta. It evaluates Rosetta's performance based on how similar the distributions are before and after the relax step, where more similar is better. This benchmark was used to train Rosetta's current energy function (beta_nov16). We used a set of 54 structures withheld from training to evaluate its performance.

Based on Rosetta's propensity to introduce large clashes during design, we expected it would introduce elevated levels of clashing when relaxing crystal structures in the benchmark. Indeed, structures relaxed with the beta_nov16 energy function had elevated levels of clashing for several atom pairs, most of which involved at least one carbon atom from a nonpolar sidechain. Fig 4A shows this pattern for two example atom pairs that are highly abundant in the structures (pairs of methyl carbons or pairs of methyl carbons and backbone oxygens), mirroring patterns seen above in designs. Fig 4B quantifies levels of clashing across a wider array of atom pairs. Since hydrogens are not always experimentally resolved in these structures, we focused our analysis on heavier atoms. In all, these results provide a clear example of beta_nov16's clashing bias in a high-quality benchmark.

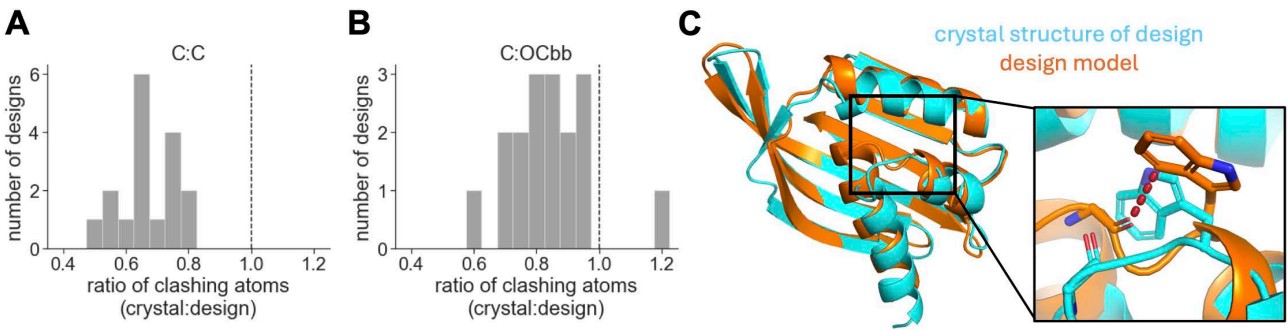

**Fig 3. Crystal structures have lower levels of clashing than corresponding design models. A)** The ratio of the number of carbon-carbon atom pairs from nonpolar sidechains that are clashing in a crystal structure vs. its corresponding design model. The design model was first relaxed in Rosetta using the beta_nov16 energy function to allow clashes to be relieved (see *Methods*). **B)** The same as panel A, but for atom pairs involving a carbon atom from a nonpolar sidechain and a backbone oxygen atom (C:OCbb). **C)** An example of a large C:OCbb clash (see dashed red line) in a designed NTF2 (MC2_7; PDB ID 6W40) [27]. This clash is present in the design model (orange), but absent in the design's crystal structure (blue) due to a large conformational rearrangement in the region of the clash. Among C:OCbb pairs, the above clash is more extreme than clashes in the design's crystal structure and nearly all clashes in a set of 54 high-resolution crystal structures of native proteins (from the atom-pair distance-distribution benchmark; S4D and S4E Fig).

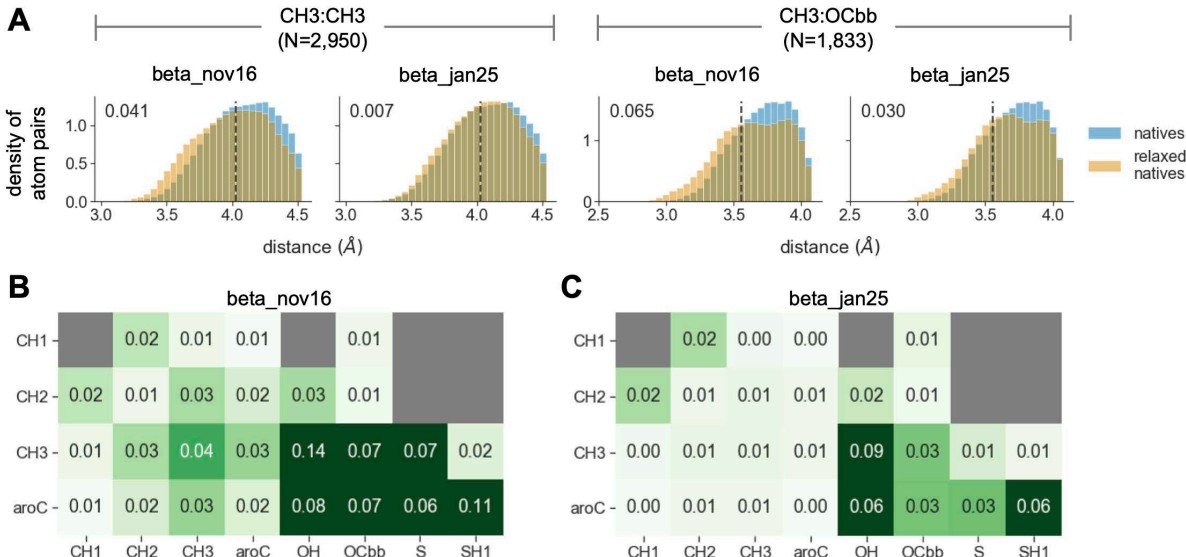

**Fig 4. Rosetta introduces steric clashes when relaxing high-resolution crystal structures of native proteins. A)** Each plot shows smoothed distributions of interatomic distances from the atom-pair distribution benchmark evaluated on crystal structures withheld from training. The left two plots show distances between pairs of methyl carbons, while the right two plots show distances between pairs of a methyl carbon and a backbone oxygen (N reports the number of instances of a given atom pair in the corresponding distribution). The dashed line shows the sum of the radii of a given atom pair. Distributions are truncated at 0.5 Å greater than this sum. Blue and orange distributions show interatomic distances before and after relaxing the structures with Rosetta, respectively, using either the beta_nov16 or beta_jan25 energy function (as indicated by the plot title). Numbers in the top left of each plot quantify the KL divergence between the blue and orange distributions, where values closer to zero indicate better agreement. **B)** Panel A reports KL divergence values for two specific atom pairs. The heatmap in this panel reports values for an array of atom pairs, when evaluating beta_nov16 on the benchmark. High values are the result of elevated levels of clashing, as in panel **A**. We do not report values for atom pairs with fewer than 100 instances in its distribution (gray boxes). **C)** Same as panel B, but evaluating beta_jan25 on the benchmark. KL divergence values are substantially closer to zero as a result of reduced clashing, as in panel **A**.

## Retraining Rosetta using high-quality benchmarks largely eliminated the clashing bias

Following the workflow from Fig 1, we next sought to use the above benchmark to retrain the Rosetta energy function with the goal of reducing its clashing bias. Previously, the parameters in the Rosetta energy function were fit using a protocol called *dualOptE* [18]. In this protocol, an ML algorithm is used to heuristically search parameter space for a set of values that maximizes the energy function's performance on an array of biophysical benchmarks.

Rosetta models van der Waals interactions using the Lennard-Jones (LJ) potential, in which the space occupied by each atom is characterized by two parameters, a radius (characterizing the volume of the atom) and a well depth (characterizing the attractive force imparted on other atoms). When beta_nov16 was trained, these parameters were fit using the atom-pair distance-distribution benchmark from above and a benchmark that involves predicting thermodynamic properties of small molecules [18]. While the distance-distribution benchmark should have identified the clashing problem, the signal was relatively weak overall as the differences were fairly subtle and only affected about 10 out of 118 atom-pair distributions.

We used *dualoptE* to retrain the energy function, using the same set of benchmarks used to train beta_nov16, but upweighting the contribution of the atom-pair distribution benchmark to the *dualoptE* loss function to amplify signal related to the clashing bias. We refit a total of 15 parameters. One controls the strength of the repulsive regime of the LJ potential (fa_rep). The others control the LJ radii and well depths of carbon and hydrogen atom types involved in the clashing problem. In training beta_nov16, LJ parameters for aliphatic carbon atom types were coupled, but we allowed them to be uncoupled in training beta_jan25. Following optimization with *dualoptE*, we manually fine-tuned parameters to further improve atom-pair distribution fits, while maintaining the same overall performance as beta_nov16 on the small

molecule-based benchmark (S5 Fig). In general, the refit parameters caused the energy function to become more repulsive (S2 Table; S6 Fig). We call the resulting energy function beta_jan25.

To test whether these changes reduced Rosetta's clashing bias, we assessed the performance of beta_jan25 on the atom-pair distribution benchmark using the set of 54 crystal structures withheld from training. The bias was largely eliminated. For instance, for the two atom pairs highlighted in Fig 4A, relaxing structures with the beta_jan25 energy function induced less clashing and resulted in more native-like distributions compared to relaxing structures with beta_nov16. This trend held for nearly all atom pairs showing the clashing bias (compare Fig 4B to Fig 4C). The refitting did not completely eliminate the clashing bias. Nonetheless, the above improvements contribute to beta_jan25 having higher overall performance on the benchmark (Table 1).

We saw similar results when we re-designed high-resolution crystal structures, rather than relaxing them, though clashing is more pronounced in design for both energy functions (S7 Fig). We did not find evidence that the increased repulsion in beta_jan25 results in *underpacking* in design. Instead, we saw low levels of residual overpacking.

In addition to validating beta_jan25's performance on the atom-pair distribution benchmark, we also validated its performance on benchmarks related to predicting protein-protein interfaces and predicting the ΔΔG of mutations on protein folding or protein-protein binding (Table 1). We chose these benchmarks for two reasons: these are areas where deep learning has had difficulty, and areas expected to be sensitive to changes in the LJ potential. The interface-prediction benchmark was used in training, but we tested the energy functions using a set of 59 structures withheld from training. beta_jan25's performance was significantly higher than beta_nov16's on this benchmark, though by a modest amount (Table 1; S8 Fig). The ΔΔG benchmarks were not used in training beta_jan25. For each of these benchmarks, we quantified the ability of both beta_nov16 and beta_jan25 to correctly classify mutations into categories of stabilizing, neutral, or destabilizing. The two energy functions had similar accuracies (Table 1). In all, these results indicate the refitting procedure substantially reduced the energy function's clashing bias, while maintaining or slightly increasing its performance on the above benchmarks. This marks the completion of the workflow from Fig 1.

## Both energy functions prefer design models to corresponding crystal structures

Since crystal structures of designs have fewer clashes than corresponding design models, we hypothesized that beta_jan25 would have a greater energetic preference for the crystal structures. To test this hypothesis, we used beta_nov16

**Table 1. Performance of energy functions on benchmarks using data withheld from training.**

| benchmark | beta_nov16 | beta_jan25 |
|---|---|---|
| atom-pair distribution[a] | 0.022 | **0.018*** |
| interface prediction[b] | 0.64 | **0.67*** |
| monomer ΔΔG prediction[c] | 0.55 | 0.55 |
| interface ΔΔG prediction[c] | 0.64 | **0.66** |

[a]The average KL divergence value across all atom pairs, weighted by the square root of the number of observations of each pair. This average is significantly lower for beta_jan25 compared to beta_nov16 according to a randomization test (*p = 0.002; see Methods). [b]Boltzmann-weighted probability of observing near-native structures in the protein-protein interface prediction benchmark, averaged across all 59 interfaces in the benchmark. The score is significantly higher for beta_jan25 compared to beta_nov16 according to a randomization test (*p < 0.001; see Methods). [c]The fraction of mutations that were correctly classified as stabilizing (<-1 kcal/mol), destabilizing (>1 kcal/mol), or neutral (between -1 and 1 kcal/mol). These tests involved a total of 735 mutations for monomers and 633 mutations for interfaces. For each test, there were small differences in classification accuracy between energy functions, but these differences were not significant according to McNemar's test (p = 0.78 for monomers and p = 0.50 for interfaces; see Methods). All values were rounded to the displayed number of significant figures.

or beta_jan25 to separately relax each design model and crystal structure, allowing each to reach a local minimum in the energy function's energy landscape (S9A and S9B Fig). Interestingly, both energy functions tended to strongly prefer designs over crystals, both doing so by an average of ~0.1 kcal/mol per residue (Figs 5A, S9E and S9F; see "total energy"). Thus, the results did not support our hypothesis.

Investigating the basis for this preference, we found that it was driven by terms modeling electrostatic interactions and hydrogen bonding, and to a lesser extent by terms modeling LJ interactions (Fig 5A). These attractive energies were opposed by terms that model solvation energies, including the energetic cost of desolvating polar atoms (Fig 5A). However, the attractive energies more than counterbalanced this cost. Recomputing energies without sidechain atoms largely eliminated the preference of the attractive terms for design models (S9D Fig), indicating sidechains largely drove the trend. Indeed, we found many examples where sidechain polar interactions were less optimal in relaxed crystal structures (e.g., Fig 5B). In all, these results suggest that the preference for design models is largely driven by polar interactions involving sidechain atoms. Since beta_nov16 and beta_jan25 are nearly identical in how they model polar interactions (the only difference being a slightly more repulsive LJ term), it makes sense that both energy functions would have similar preferences for design models.

The above preference for design models could suggest a second failure mode common to both beta_nov16 and beta_jan25: that both energy functions model certain attractive polar-interaction energies too strongly. We do not attempt to address this potential failure mode, but highlight it to help guide future energy-function development.

## Discussion

*De novo* protein design gives an energy function a large amount of freedom to optimize sequences and structures according to the energy function's view of reality. When designs are experimentally tested, common failure modes can suggest underlying inaccuracies in the energy function. In this paper, we established a framework to use such failure modes to guide energy-function development. Specifically, we identified a common failure mode (steric clashing) in proteins designed using Rosetta, identified examples of this failure mode in a high-quality benchmark, and then used this benchmark to help retrain the Rosetta energy function. We make the updated energy function (beta_jan25) available in the Rosetta software package.

This work contributes to the field in multiple ways. First, it shows that DMS of unstable protein designs can reveal putative failure modes. Most DMS experiments are performed on proteins that are already functional. Our DMS experiments were performed on protein designs with little-to-no initial folding stability according to the high-throughput stability assay. For most designs, we

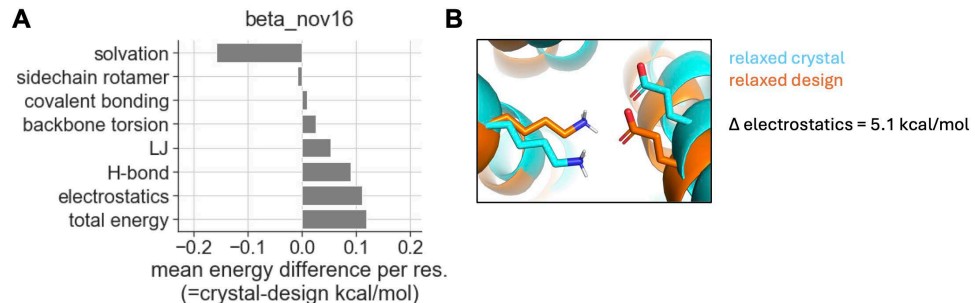

**Fig 5. Energy differences between design models and corresponding crystal structures. A)** Energy differences between design-crystal pairs relaxed with the beta_jan25 energy function. We performed six independent relax replicates, and then selected the replicate with the lowest total energy for computing energy differences. Plotted differences are averaged over all design-crystal pairs. Each bar shows differences computed for a given set of energy terms: "total energy" includes all energy terms, while the other bars include energetically related subsets of terms that are non-overlapping between categories (see *Methods*). **B)** An example of a polar interaction between sidechains that is present in the relaxed design model (orange), but less optimal in the relaxed crystal structure (blue; D_3_212, residues K22 and E220). The electrostatic energy difference for this particular pair of residues is 5.1 kcal/mol (=crystal-design) as computed using the beta_nov16 energy function.

identified at least one amino-acid mutation that dramatically improved the design's stability. These rescue mutations formed patterns that suggested multiple distinct failure modes, even for the same design. It is unclear whether most of the unstable miniprotein designs from Rocklin et al. would have been rescued by single mutations, or just the "outlier" designs we selected.

Second, to our knowledge, this is the first study that systematically examined a large and diverse set of design models and corresponding crystal structures in search of common failure modes driving structural differences. This comparison helped identify the putative failure mode related to clashing, and also suggested another related to the strength of attractive polar-interaction energies. These results point to the value of experimentally determining structures of designs, including instances where the structure is substantially different from the design model.

Third, our work explores a new paradigm for developing macromolecular models: learning from *de novo* protein design. Initially, we imagined using data on designs to directly retrain the Rosetta energy function, helping to augment the limited diversity of available training data. This direct approach could still be feasible. However, we found success in an indirect approach that helped expose a failure mode in a benchmark that was already used in training. Signal for this failure mode was weak in the benchmark. But, once we became aware of it, we were able to upweight the benchmark in *dualoptE* and largely eliminate signal of the failure mode in a subsequent round of optimization. This general approach of indirectly learning from designs could be applied to any macromolecular model used in design, including deep-learning models.

Fourth, we identified a systematic bias wherein proteins that are relaxed or designed with the beta_nov16 energy function tend to have elevated levels of clashing compared to crystal structures. It is possible that designed clashes interfere with proteins taking on designed conformations in reality. Relatedly, in a recent study that used Rosetta to design *de novo* enzymes, the authors identified multiple mutations that improved the designs, including a large-to-small mutation that relieved a steric clash in the design model [34]. This result provides additional evidence that designed clashes can be problematic.

Avoiding overpacking could be a general challenge for macromolecular models trained on protein structures. The repulsive component of the LJ potential scales very steeply with distance ($1/d^{12}$), such that even slight overpacking can lead to large energy costs. However, signal of overpacking may be subtle in benchmarks, as we saw in the benchmark that explicitly examined interatomic distances. In training beta_jan25, amplifying this signal was effective at greatly reducing the overpacking problem. Despite this improvement, the problem persists at a low level for some atom pairs. Future work could examine ways to completely eliminate the problem. In this paper, we focused on refitting a small number of parameters in context of the 12–6 LJ potential. But, future work could explore more dramatic refitting schemes, including alternative functional forms for modeling van der Waals interactions [35].

In all, this study provides a template for using experimental testing of designs to guide continued optimization of molecular models in the age of high-throughput data-driven learning.

## Methods

### Data and code availability

See https://github.com/Haddox/design_guided_optE for input data, output data, and code. The Rosetta source code is freely available for academic use in the Rosetta software suite at https://rosettacommons.org/. We have added the beta_jan25 energy function to the Rosetta source code. The energy function can be invoked using the "-beta_jan25" flag (see https://docs.rosettacommons.org/docs/latest/full-options-list). To implement the precise energy function used in the dualoptE protocol, also include the flag "-score::count_pair_hybrid false".

### Identification of unstable miniprotein designs for deep mutational scanning

We examined miniprotein designs from Rocklin et al. [19], focusing on designs with HHH, EHEE, or EEHEE topologies. Rocklin et al. trained machine-learning models to predict the stability scores of these designs using an array of biophysical features (e.g., buried nonpolar surface area, number of buried unsatisfied polar atoms, etc.), including Rosetta energies.

Building on this work, we computed an expanded set of features using an updated pipeline (see https://github.com/Haddox/score_monomeric_designs). For each miniprotein topology, we then trained a random-forest regression model to use those features to predict a design's stability score (see S1 Text for details on model training). S1A Fig shows the correlation between predicted and observed stability scores for each topology. We identified 21 outlier designs that were unstable according to the high-throughput stability assay from Rocklin et al. (stability score $< 0.3$, $EC_{50} < 1.5$, and an $EC_{50}$ 95% confidence interval $< 2$), but were predicted by the random-forest model to have a stability score $> 1$. To compute the Rosetta energy of each design (Figs 2A, S1A), we used the beta_nov16 energy function to first relax the design with Rosetta's *FastRelax* protocol and then evaluate its energy. See https://github.com/Haddox/design_guided_optE/tree/main/id_outlier_designs and https://zenodo.org/records/17330114 for files giving Rosetta energies, biophysical features, and ML-predicted stability scores for each design.

## Deep mutational scanning of unstable miniprotein designs

As part of the Rocklin et al. study that devised the high-throughput stability assay [19], they used the assay to measure the effects of all single amino-acid mutations to several individual designs, thus performing deep mutational scans on these designs. We used the same approach to measure the effects of all single amino-acid mutations to the 21 outlier designs, with the following modifications. We used an updated version of the high-throughput stability assay described in Maguire et al. [36], which involves using a more protease-resistant yeast-display vector that expands the dynamic range of the assay. As in Rocklin et al., we ordered saturation mutagenesis libraries as pools of oligonucleotides that encode each unmutated design as well as each single amino-acid variant of each design. We ordered these libraries in two larger libraries, with one library containing all sequence variants for 19 designs and the other containing all sequence variants for the other 2 designs. As in Maguire et al., in the first library, we also included sequences for 338 miniproteins from Rocklin et al. to assess experimental reproducibility. For the 19 designs in the first library, we performed biological duplicates of the high-throughput stability assay starting from cloning of the oligonucleotide libraries into yeast-display plasmids. For the two designs in the second library, we only performed one replicate. We separately performed the assay with the same two proteases at the same concentrations as in Maguire et al. For each protease, we used the resulting deep-sequencing data to infer $EC_{50}$ values and stability scores as described in Rocklin et al., aggregating stability scores between the two proteases by taking the minimum value for a given design.

In analyzing the results of the DMS experiments, we found that most variants were well-represented in the library before selection (S1B Fig). Stability scores were correlated between biological replicates (S1C Fig; all other figures show stability scores averaged between replicates unless otherwise noted). Among the 21 outlier miniproteins and the additional 338 miniproteins with stability scores measured in both our study and Rocklin et al., the stability scores were correlated between studies, though our measurements spanned a larger dynamic range due to the updated yeast-display vector (S2D Fig). See https://github.com/Haddox/design_guided_optE/tree/main/rescue_dms for files giving deep-sequencing counts, protease $EC_{50}$ values, and stability scores from each experiment, as well as oligonucleotide sequences.

## Comparison of design models and corresponding crystal structures

For each of the crystal structures of designs listed in S1 Table, we compared the crystal structure with its original computational design model in two main ways described below. For a few crystal-design pairs, a small fraction of sites were unresolved in the crystal structure. A few pairs also differed in their underlying amino-acid sequence at a small number of sites, mostly at the protein termini. To help enable a head-to-head comparison of these design-crystal pairs, we made a modified version of each structure so that both structures in the pair had the exact same set of sites and amino-acid sequence. This involved deleting experimentally unresolved sites in the design model, as well as deleting nonidentical sites at protein termini. For two design models (WSHC6 and 5L6HC3_1), we used Rosetta to make a small number of

mutations to the design structure so that it matched the sequence of the crystal structure. All mutations were at residues on the protein surface.

The first way we compared design-crystal pairs was by analyzing their levels of clashing. To do so, we first used Rosetta's *FastRelax* protocol to relax each design model in Cartesian space using the beta_nov16 energy function. Design models were often generated in torsional space. We relaxed structures in Cartesian space to give the energy function increased freedom to potentially relieve steric clashes. We compared these relaxed design models to crystal structures that had not been relaxed.

The second way we compared design-crystal pairs was by analyzing their Rosetta energies. For this analysis, we used Rosetta's *FastRelax* protocol to relax each design model and crystal structure in Cartesian space using either the beta_nov16 or beta_jan25 energy function. We then computed the energies of the relaxed structures using the same energy function used in the relax step. We performed 6 independent replicates of the relax protocol and then selected the replicate structure with the lowest total energy for downstream comparison. To assess the contribution of sidechain atoms to the energies, we used Rosetta to convert the sequence of the relaxed structures to poly-glycine, which effectively replaced all sidechain atoms with a single hydrogen atom while keeping all backbone atoms fixed in space, and then recomputed energies. In Fig 5A, we grouped score terms from Rosetta's energy function as follows: total energy (all terms), LJ (fa_atr, fa_rep, fa_intra_atr_xover4, fa_intra_rep_xover4), solvation (fa_sol, lk_ball, lk_ball_iso, lk_ball_bridge, lk_ball_bridge_uncpl, fa_intra_sol_xover4), electrostatics (fa_elec, fa_intra_elec), H-bond (hbond_sc, hbond_bb_sc, hbond_lr_bb, hbond_sr_bb), covalent bonding (cart_bonded), backbone torsion (omega, p_aa_pp, rama_prepro), sidechain rotamer (fa_dun_rot, fa_dun_dev, fa_dun_semi, hxl_tors).

See https://github.com/Haddox/design_guided_optE/tree/main/designs_and_xtals for the structures of each design and crystal pair before and after modifying them to have the same sequence and before and after relaxing the modified structures, as well as code for performing the protocols described in this section, as well as associated output files and plots.

## Atom-pair distance-distribution benchmark

We performed this benchmark as described in Park et al. [18] with the following modifications. First, we used a slightly modified the atom-typing scheme, such that distance distributions are computed separately for carbon atom types from polar vs. nonpolar sidechains. In evaluating the performance of energy functions on the 54 high-resolution crystal structures used for validation, we performed a longer relax protocol than the one used in training. See https://github.com/Haddox/design_guided_optE/tree/main/atom_pair_benchmark for the 78 and 54 high-resolution crystal structures used in training and validation, the code for relaxing structures and computing atom-pair distance distributions, and output files with the resulting distributions.

## Training the beta_jan25 energy function

We trained beta_jan25 using the same *dualoptE* protocol with the same set of benchmarks that Park et al. [18] used to train opt-nov15, with the following modifications. We initialized all parameters at their values in the beta_nov16 energy function. We only refit the 15 parameters listed in S2 Table. The *dualoptE* loss function is a weighted sum of scores across benchmarks. We increased the weight of the atom-pair distance-distribution benchmark by 10-fold. Although this is a large increase, beta_nov16's unweighted score on this benchmark is about 10-fold lower than other benchmarks, so the upweighting did not cause this benchmark to contribute substantially more than others to the loss function. We also used an updated version of the interface-prediction benchmark (described in the next section).

After running the automated *dualoptE* protocol, we manually fine tuned parameters to further improve performance on the distance-distribution benchmark used in training. That is because some atom pairs still showed low residual levels of clashing after the *dualoptE* protocol. We found that manually increasing radii by a small amount helped reduce this residual clashing, slightly increasing performance on the atom-pair distribution benchmark without decreasing performance

in the benchmarks. The residual clashing after *dualoptE* may have been because signal of this residual is weak in the benchmark's loss function when summed over all atom pairs.

### Interface prediction benchmark

We performed this benchmark as described in Park et al. [18] with the following modifications. We trained the beta_jan25 energy function using a set of 65 crystal structures, and then validated its performance using 62 crystal structures withheld from training. The benchmark focuses on crystal structures of interfaces between two protein chains. For each structure, Park et al. computationally generated 1,000 decoys where the two chains were rigidly docked in non-native orientations, used Rosetta to relax the decoys, and then evaluated Rosetta's ability to discriminate between near-native and non-native decoys.

One modification we made was to sample more decoys. For each decoy from Park et al., we randomly jittered one chain relative to the other, doing so many times to create a cloud of closely related decoys. Specifically, each jitter involved rigidly translating the chain between 0–2 Angstroms in a random direction in 3D space, coupled with rigidly rotating the chain between 0–10 degrees about a random axis. From each cloud, we chose the five decoys with the most favorable Rosetta energies, when evaluated using the "beta_soft" energy function, which downweights the repulsive term of the Lennard-Jones potential to allow low levels of clashing that could be relieved during a relax protocol. This resulted in a total of 5,000 decoys per crystal structure.

Another modification we made was to allow Rosetta to relax both backbone and sidechain atoms of native and decoy structures. In the original benchmark, Rosetta was only allowed to relax sidechain atoms (backbone atoms were kept fixed). Specifically, our protocol involved the following steps. First, we used the beta_nov16 energy function to relax each structure using Rosetta's *FastRelax* protocol, allowing all residues to be relaxed. Second, we used either the beta_nov16 or beta_jan25 energy function to perform an additional relax step, this time only relaxing residues near the protein-protein interface, defined as residues within 12 Angstroms of the opposite chain. For each structure, we independently performed the second relax step with three different inputs: both chains in the complex (chains A and B), just chain A, or just chain B. We used the Rosetta energies of the outputs to compute the $\Delta G$ of binding for a given structure, where $\Delta G = E_{complex} - (E_{chain\ A} + E_{chain\ B})$. This resulted in an energy landscape of relaxed native and decoy structures for each of the starting crystal structures. We evaluated the performance of each energy function by analyzing the landscape generated by using that energy function in the second relax step from above. To do so, we binned structures by their $C_\alpha$ RMSD to the starting crystal structure, using a bin size of 2 Angstroms, and then uniformly selected the structures with the lowest $\Delta G$ values from each bin until we reached a total of 150 structures. We then quantified Roestta's ability to discriminate between near-native and non-native decoys in each landscape using a Boltzmann-weighted discrimination score, as described in Park et al.

Carrying out the above relax protocols on each of the thousands of decoys from above required a large amount of compute time. To help make this task feasible, we used Rosetta@home, which is a distributed computing platform that runs Rosetta protocols on computers of public volunteers. While useful, some of the relax protocols that we submitted stochastically failed, which is expected for this type of platform, resulting in dropout. For the protocols that succeeded, Rosetta generated one to several replicate output structures per input structure. We took all replicate output structures from the first relax step from above (the all-residue relax) and used them as individual inputs for the second relax step (the interface-residue relax). For each input structure to the second relax step, we only analyzed the resulting output data if the complex, chain-A, and chain-B relax protocols each produced at least two replicates. We defined $E_{complex}$, $E_{chain\ A}$, and $E_{chain\ B}$ to be the minimum energy across all replicates of the complex, chain-A, or chain-B relax protocols, respectively. Further, for a given input structure to the second relax step, we discarded the resulting output data if they failed to pass the following quality-control filters:

1. We discarded the output data if the minimum and median energies among all replicates of either the complex, chain-A, or chain-B jobs differed by >0.1 energy units per residue. We expect replicates to have small energy differences due to stochasticity in the protocol. Large differences point to instability in the protocol for a given input.

2. We discarded the output data if either the $E_{chain\,A}$ or $E_{chain\,B}$ value for the structure was > 0.1 energy units per residue higher than the median $E_{chain\,A}$ or $E_{chain\,B}$ values among all structures in the same landscape. Without this filter, a small number of structures had very negative ΔG values merely because the associated $E_{chain\,A}$ or $E_{chain\,B}$ values were very high relative to the other decoys in the landscape. We expect some deviation from the median due to stochasticity in the relax protocol. But, very high values that strongly deviate from the median indicate the relax protocol may have stochastically failed.

For >90% of the 62 crystal structures in the validation set, < 5% of decoys were filtered out by the first filter, and <5% by the second filter. For one crystal structure, we discarded >15% of decoys, so we chose to entirely exclude this structure from our analysis. We ultimately obtained ΔG estimates for both beta_nov16 and beta_jan25 for a median of 5,629 decoys per crystal structure and an interquartile range of 3,642–7,204 decoys. This number was < 1,000 for 3 of the 62 crystal structures, which we excluded from our analysis due to high dropout.

When using *dualoptE* to train the beta_jan25 energy function, we included this benchmark, but adapted it to make it fast enough to use in *dualoptE*. As input, we used structures that had already been relaxed with the beta_nov16 energy function, choosing 150 low-energy structures per landscape, uniformly sampled across RMSD values, as described above. At each step of optimization, for each of the above structures, we used the energy function, in conjunction with Rosetta's *PackRotamersMover* protocol, to optimize sidechain rotamers of interface residues of the complex, just chain A, or just chain B. We then computed ΔG values and discrimination scores as described above. See https://github.com/Haddox/design_guided_optE/tree/main/interface_prediction for code and output score files. See https://zenodo.org/records/17330114 for the decoys used in *dualoptE* (150 per landscape) and https://files.ipd.uw.edu/pub/2025_optE/all_boinc_out_files.tar.xz to download all output decoys from the first relax step from above (all residues relaxed with beta_nov16).

## Benchmarks on predicting mutational effects on free energies

One benchmark involved predicting effects of mutations on free energies of protein folding. We performed this benchmark as described in Frenz et al. [9], with the following modifications. We focused on the balanced set of 768 mutations from "Data Sheet 2.CSV" from Frenz et al. To predict the ΔΔG of a given mutation on a given structure, we first stripped that structure's PDB file to only include the chain in which the mutation occurred. Next, we used Rosetta's *FastRelax* protocol to perform a constrained relax of the structure in Cartesian space, performing five independent relax replicates. We retained the replicate with the lowest Rosetta energy. Using this relaxed structure as input, we then used Rosetta's Cartesian ΔΔG protocol [9,18] to compute the ΔΔG of the mutation in context of the relaxed structure, performing five iterations of the protocol and averaging ΔΔG values across iterations to obtain a final ΔΔG estimate. We performed the relax and ΔΔG protocols separately for each energy function and compared the predicted values to experimentally measured ΔΔG values reported in the CSV from Frenz et al. We obtained estimated mutational effects for 735 mutations for each energy function, a small number of input PDBs having been incompatible with our pipeline. In Table 1, we used McNemar's test to test for a significant difference in the classification accuracy between energy functions. The input to this test was a 2x2 contingency table reporting the number of mutations correctly classified by both energy functions, the numbers correctly classified by just beta_nov16 or just beta_jan25, and the number incorrectly classified by both. See https://github.com/Haddox/design_guided_optE/tree/main/monomer_ddg for the specific commands we used to perform these steps, as well as estimated ΔΔG values.

The other benchmark involved predicting effects of mutations on free energies of binding of protein-protein complexes. We used experimentally measured data from the SKEMPI v2.0 database [37], downloading these data and corresponding PDBs of the interfaces from https://life.bsc.es/pid/skempi2/database/index; 08.06.2018 file versions. Specifically, the data reported the $K_d$ of binding of both the wildtype and mutant complexes, along with the temperature of the assay in

Kelvin (T). We converted $K_d$ values to ΔG values in units of kcal/mol using the equation $\Delta G = 1.987e\text{-}3 * T * \ln(K_d)$. We then obtained ΔΔG values as $\Delta\Delta G = \Delta G_{mut} - \Delta G_{wt}$. We downsampled the 6,185 mutations with data to a set of 633 mutations split roughly evenly between four mutational categories: polar-to-polar, polar-to-nonpolar, nonpolar-to-polar, and nonpolar-to-nonpolar amino-acid mutations. We randomly selected mutations in each category among structures with at least 10 ΔΔG measurements, and among mutations that were both measured in a single mutant variant context and for which the assay temperature was known (not assumed). For practical reasons related to Rosetta's Cartesian ΔΔG protocol, we focused on a set of 62 structures for which an alphabetically sorted list of single-letter chain names from the PDB file could be divided into two pieces giving the chains on either side of the interface, with at least one side having a single chain. As above, to predict the ΔΔG of a given mutation to a given complex, we first used Rosetta's *FastRelax* protocol to perform a constrained relax of the structure in Cartesian space, performing five replicates and retaining the replicate with the lowest total energy, and then used this structure as input to Rosetta's Cartesian ΔΔG protocol, performing three iterations of the protocol. The output consisted of computed ΔG of binding values for both the wildtype and mutant complexes. For each complex, we took the average ΔG across the three iterations, and then computed the final ΔΔG estimate as $\Delta G_{mut} - \Delta G_{wt}$. See https://github.com/Haddox/design_guided_optE/tree/main/interface_ddg for specific commands used to perform these steps, as well as estimated ΔΔG values.

## Randomization testing

In Table 1, we used randomization to test whether the scores from the atom-pair distribution benchmark and interface-prediction benchmark were significantly better for beta_jan25 compared to beta_nov16. We did so using the following approach. For the interface-prediction benchmark, each energy function's score in Table 1 is an average of the individual scores for each of the 59 interfaces in the benchmark. beta_jan25's average score is higher (better) than beta_nov16's average score by 0.03. Various sources of noise affect scores of individual interfaces, such that nonzero differences between energy functions could arise by chance. To test whether the observed difference is significantly greater than zero, we generated a null distribution by iterating over each of the 59 interfaces, randomly shuffling the scores associated with each energy function for that interface (such that half the time the scores were assigned to the same energy function as before and half the time they were swapped), and then computing the difference in average scores between energy functions (= beta_jan25 - beta_nov16). We independently repeated this randomization protocol 1,000 times. None of the differences in average score from the null distribution were greater than or equal to the observed difference in average score, indicating the observed difference is significantly greater than zero with $p \leq 0.001$.

In the case of the atom-pair distribution benchmark, each energy function's score in Table 1 is a weighted average of individual scores for a large number of atom pairs. beta_jan25's weighted average is lower (better) than beta_nov16's weighted average, with a difference of -0.004 (= beta_jan25 - beta_nov16). Similar to above, we tested whether this difference was significantly less than zero by randomly shuffling scores associated with each energy function for a given atom pair and then recomputing the difference in weighted averages between energy functions, independently performing this randomization step 1,000 independent times. Only 2/1,000 values from the resulting null distribution were less than or equal to the observed difference, indicating the observed difference is significantly less than zero with $p = 0.002$.

## Supporting information

**S1 Text. A file with supplemental text on random forest model training and data-driven identification of outlier designs.**
(DOCX)

**S1 Table. List of crystal structures of designs analyzed in this study.** See https://github.com/Haddox/design_guided_optE/tree/main/designs_and_xtals for PDBs for each design model and its corresponding crystal structure, named

according to this table. Most designs in the table have helical topologies, though a few have mixed *α/β* topologies, and one is a *β* barrel.
(XLSX)

**S2 Table. Differences in parameter values between beta_nov16 and beta_jan25.**
(XLSX)

**S1 Fig. Additional results from deep mutational scanning of outlier designs. A)** The top row of plots are similar to Fig 2A, but show data for all three miniprotein topologies from Rocklin et al. that we analyzed in this study (HHH, EHEE, and EEHEE; letters indicate the order of secondary-structure elements in a topology, with H = helix and E = strand). The bottom row of plots are the same as the top row, but show stabilities predicted by ML models instead of Rosetta (see Methods). Red dots show all 21 outlier designs selected for DMS, which span all three topologies. **B)** Histograms show the number of deep-sequencing counts of all single amino-acid variants in starting DMS libraries after transforming them into yeast cells. As described in the *Methods*, the 21 DMS libraries were ordered and tested as part of two larger libraries. The first of these libraries was tested in duplicate. Each plot shows data for a single replicate of a given library. Distributions are capped at 400 counts. **C)** Correlation of stability scores between the two biological replicates of library 1, which were generated from independent transformations of unselected plasmid libraries into yeast. In the scatter plot (left), each dot corresponds with a unique variant in the library. The 2D histogram (right) shows that most variants had low stability scores near zero in each replicate. **D)** Correlation of stability scores between this study and Rocklin et al. for all outlier designs, as well as 338 "ladder" miniproteins from Rocklin et al. (see Methods). Red dots show outlier designs from library 1, blue dots show outlier designs from library 2, and gray dots show ladder miniproteins. *R* is the Pearson correlation coefficient. The higher dynamic range of values in this study comes from experimental modifications that made the yeast surface-display scaffolding proteins less sensitive to proteolysis (see Methods). **E)** The same as Fig 2C, but with the y-axis scaled in log space for increased visibility of the tails of the distribution. **F)** The same as panel E, but now showing mutational effects pooled across the 7 outlier designs with a stability score >0.5, as measured in this study.
(TIFF)

**S2 Fig. Stability scores of all single amino-acid variants from deep mutational scanning of two of the unstable outlier designs. A)** Data for HHH_rd4_0216 (labeled with an orange triangle in Fig 2G). **B)** Data for HHH_rd4_0628 (labeled with a purple circle in Fig 2G). Each box shows the stability score of a variant with a given amino-acid mutation (y-axis) at a given site (x-axis). Boxes with dots correspond to the unmutated design and show its stability score (the horizontal black line on the color bar also shows the unmutated design's score). Numbers show stability scores for variants with scores of at least 1.0. We were unable to measure stability scores for a small number of variants (black boxes). Heatmaps of all 21 outlier designs are available at https://github.com/Haddox/design_guided_optE/tree/main/rescue_dms/heatmaps.
(TIFF)

**S3 Fig. Levels of clashing in miniprotein designs exceed levels seen in high-resolution crystal structures of native proteins.** Each plot shows smoothed distributions of interatomic distances for a given atom pair (columns) within a given set of proteins (rows). Blue distributions show distances observed in 78 high-resolution crystal structures, showing data for all instances of a given atom pair within a distance cutoff of 0.5 *Å* greater than the sum of the radii of these atoms. Orange distributions show distances observed in a given set of designs from Rocklin et al. (from design rounds 3 and 4), with rows of plots separating designs by topology (HHH, EHEE, EEHEE, HEEH) and whether the designs were stable (stability score > 1.0) or unstable (stability score < 1.0). Vertical dashed lines show the sum of the van der Waals radii of the atom pair. Interatomic distances to the left of this line correspond to "clashes". Columns show data for different atom pairs

from hydrophobic sidechains, including pairs of hydrogens (Hapo:Hapo), hydrogens and methyl carbons (CH3:Hapo), or methyl carbons (CH3:CH3). In each plot, designs show higher levels of clashing than the crystal structures. (TIFF)

**S4 Fig. Clashing in design models compared to corresponding crystal structures. A)** An example of a large clash between carbon atoms from nonpolar sidechains (see dashed red line) in a designed heterodimer (DHD127; PDB ID 6DLM). The clash is present in the design model (orange), but absent in the crystal structure (blue) due to a conformational rearrangement in the region of the clash, where one of the clashing sidechains shifts up to occupy space previously occupied by a loop. The clash is more extreme than most clashes in the design's crystal structure and in a set of 54 crystal structures of native proteins, as shown in the next two panels. **B)** Histograms show the distribution of interatomic distances between all pairs of carbon atoms from nonpolar sidechains (C:C) in the DHD127 crystal structure (blue distribution) or design model (orange distribution) within a distance cutoff of 0.5 Å greater than the sum of the radii of these atoms (the dashed line shows this sum). The red arrow shows the distance of the clashing atom pair highlighted in panel A. **C)** Same as panel B, but the blue distribution shows interatomic distances from a set of 54 high-resolution crystal structures of native proteins. The two distributions are plotted as normalized densities to make them comparable. **D and E)** These three panels are similar to panels B and C, but show data for a designed NTF2 from Fig 3C, and examine clashing between pairs of carbon atoms from nonpolar sidechains and backbone oxygen atoms (C:OCbb). The red arrow shows the distance of the clashing atom pair highlighted in Fig 3C. See https://github.com/Haddox/design_guided_optE/tree/main/designs_and_xtals/compute_interatomic_distances/distance_distribution_plots for similar plots for all design-crystal pairs. (TIFF)

**S5 Fig. Energy-function performance on the benchmark for predicting thermodynamic properties of small molecules.** For a panel of small molecules, the benchmark uses a liquid-simulation framework to predict each molecule's heat of vaporization and density, given an input energy function. We performed this benchmark as described in Park et al. **A)** Fractional error in predicted values for heat of vaporization (Hvap) and density. Each bar shows the fractional error for a given energy function (hue) on a specific small molecule (x-axis). Values are averaged over predictions from four replicate simulations for beta_nov16 or three replicate simulations for beta_jan25. (NMA = N-methylacetamide, NMF = N-methylformamide, DME = dimethyl ether). **B)** The average fractional error of each energy function in predicting the properties from panel A. (TIFF)

**S6 Fig. Differences in LJ energy landscapes between energy functions for select atom pairs.** Each plot maps the LJ energy landscape for a given atom pair (see plot title) evaluated with a specific energy function (see legend). Landscapes are shifted to the right for beta_jan25 compared to beta_nov16, consistent with beta_jan25 being more repulsive at shorter distances. Shifts are subtle – on the order of tenths of Angstroms – but they are effective at reducing clashing in the distance-distribution benchmark, where overpacking is also on the order of tenths of Angstrom (Fig 4A), including for atom pairs like CH3:CH3 and CH3:OCbb for which clashing is particularly pronounced (Fig 4). The shift is most pronounced for CH3:CH3, which is expected since CH3 had the largest increase in LJ radius upon refitting (S2 Table). (TIFF)

**S7 Fig. Changes in atom-pair distance distributions upon re-designing high-resolution crystal structures of native proteins with Rosetta.** This figure is similar to Fig 4A, but shows the results of re-designing high-resolution crystal structures, rather than relaxing them. Each panel (A-D) shows data for a different atom pair (see panel titles). We performed this re-design test using 20 of the 54 crystal structures used for validation. Similar to the relax test, there is less clashing in structures re-designed with beta_jan25 compared to those re-designed using beta_nov16. For both energy

functions, clashing is more pronounced in design than relax, which might be expected since design gives the energy function more freedom. There is still some residual clashing in structures re-designed using beta_jan25 (the orange distributions are still shifted to the left).
(TIFF)

**S8 Fig. Energy function performance in the interface-prediction benchmark.** Each dot corresponds to one of the 59 crystal structures withheld from training. For each of these crystal structures, we generated a large number of decoys with non-native interfaces, relaxed the decoys with a given energy function, and then computed the Boltzmann-weighted probabilities of observing near-native structures (see Methods), with higher probabilities indicating better performance. The x and y axes report probabilities obtained by relaxing the structures with either beta_nov16 or beta_jan25, respectively. Table 1 reports the average probability for a given energy function. The higher average for beta_jan25 comes from modest increases in probabilities for several structures (dots above the diagonal line). There are only a few structures where the probability is lower for beta_jan25 (dots below the diagonal line).
(TIFF)

**S9 Fig. Energy differences between design models and corresponding crystal structures. A)** We separately relaxed each design model and each crystal structure, performing six independent replicates of the relax protocol. This panel shows the results for one design-crystal pair (DHD131) relaxed with the beta_nov16 energy function. Each dot corresponds the output structure from a single replicate of relaxing the crystal structure (blue dots) or the design model (orange dots). The y-axis shows each structure's total energy per residue, while the x-axis shows each structure's $C_\alpha$ RMSD to the unrelaxed crystal structure. See https://github.com/Haddox/design_guided_optE/tree/main/designs_and_xtals/relax_and_scoring_protocols/energy_landscape_plots for similar plots for all design-crystal pairs. For a given pair, we selected the lowest-energy replicate from each color category to compute energy differences. **B)** The $C_\alpha$ RMSD of relaxed design models (orange) and crystal structures (blue) to the unrelaxed crystal structure of the corresponding design. This plot shows data across all design-crystal pairs. **C)** Distributions of energy differences of design-crystal pairs. Vertical dashed lines show the mean of each distribution. **D)** For each relaxed design model and each relaxed crystal structure, we created a version of the structure where all side-chain atoms were replaced with a single hydrogen atom (converting the sequence to poly-glycine while keeping all backbone atoms fixed in space). This panel shows energy differences computed using the unmodified structures with all side-chain atoms intact (full pose) compared with the poly-glycine structures (poly-Gly pose), with one dot for each design-crystal pair. Many of the energy differences are closer to zero for the poly-glycine structures. **E)** The same as Fig 5A, but for design-crystal pairs relaxed with the beta_nov16 energy function. **F)** The total energy differences of design-crystal pairs are highly correlated between energy functions.
(TIFF)

## Acknowledgments

We thank Michael Weim, Luki Goldschmidt, Rocco Moretti, and Sergey Lyskov for support with Rosetta and Rosetta@home; members of the Baker lab and the SD2 consortium for sharing computational design models and data files and for useful discussions, especially Chris Norn and Jedediah Singer; UW BIOFAB technicians; and members of the public who donate their computer time to Rosetta@home.

## Author contributions

**Conceptualization:** Hugh Haddox, Gabriel J. Rocklin, Francis C. Motta, David Baker, Frank DiMaio.

**Formal analysis:** Hugh Haddox, Gabriel J. Rocklin, Francis C. Motta.

**Investigation:** Hugh Haddox, Gabriel J. Rocklin, Francis C. Motta, Devin Strickland, Samer F. Halabiya, Cameron Cordray.

**Methodology:** Hugh Haddox, Francis C. Motta, Devin Strickland, Hahnbeom Park, Frank DiMaio.

**Software:** Hugh Haddox, Francis C. Motta, Frank DiMaio.

**Supervision:** Eric Klavins, David Baker, Frank DiMaio.

**Writing – original draft:** Hugh Haddox, Francis C. Motta, Frank DiMaio.

**Writing – review & editing:** Hugh Haddox, Gabriel J. Rocklin, Francis C. Motta, Devin Strickland, Hahnbeom Park, Eric Klavins, David Baker, Frank DiMaio.

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
