## [Decision Letter · Decision Letter 0]

3 Feb 2026

Using experimental results of protein design to guide biomolecular energy-function development

PLOS Computational Biology

Dear Dr. Haddox,

Thank you for submitting your manuscript to PLOS Computational Biology. After careful consideration, we feel that it has merit but does not fully meet PLOS Computational Biology's publication criteria as it currently stands. Therefore, we invite you to submit a revised version of the manuscript that addresses the points raised during the review process.

We look forward to receiving your revised manuscript.

Kind regards,

Shi-Jie Chen

Academic Editor

PLOS Computational Biology

Arne Elofsson

Section Editor

PLOS Computational Biology

**Journal Requirements:**

At this stage, the following Authors/Authors require contributions: Hugh Haddox, David Baker, Cameron Cordray, Frank DiMaio, Samer Halabiya, Eric Klavins, Francis Motta, Hahnbeom Park, Gabriel Rocklin, and Devin Strickland. Please ensure that the full contributions of each author are acknowledged in the "Add/Edit/Remove Authors" section of our submission form.

3) We noticed that you used the phrase 'data not shown' in the manuscript. We do not allow these references, as the PLOS data access policy requires that all data be either published with the manuscript or made available in a publicly accessible database. Please amend the supplementary material to include the referenced data or remove the references.

5) We notice that your supplementary Figures, and Tables are included in the manuscript file. Please remove them and upload them with the file type 'Supporting Information'. Please ensure that each Supporting Information file has a legend listed in the manuscript after the references list.

7) Kindly revise your competing statement in the online submission form to align with the journal's style guidelines: 'The authors declare that there are no competing interests.'

**Reviewers' comments:**

Reviewer's Responses to Questions

**Comments to the Authors:**

Reviewer #1: The authors performed deep mutational scanning on mini-proteins designed using the Rosetta energy function, and identified "rescue" mutations that convert unstable designs into stable proteins. Through analysis of these mutations, they uncovered common "failure modes" of the energy function. Among several such modes, they focused on one involving steric clashes or overpacking in the hydrophobic core. Notably, similar overpacking artifacts were observed when applying Rosetta to refine high-resolution crystal structures. To address this, the authors reparameterized a small subset of energy function terms to better recapitulate native-like atomic contact distances. They demonstrate that the revised energy function corrects the steric clashes or overpacking in structure refinement tasks, while maintaining comparable performance on benchmarks for predicting ΔΔG values of monomer and interface mutations.

Given the widespread use of the Rosetta energy function, this study is highly relevant and delivers a practically useful update to an important tool. Nevertheless, I have the following concerns that need to be addressed before publication.

While overpacking appears as one failure mode among others in the deep mutational scanning data, it seems to represent only a minor subset of the overall design failures. Although it is reasonable to focus on this problem at first, it is essential to evaluate whether the proposed reparameterization disrupts the delicate balance between different energy terms in the original function. Such rebalancing could inadvertently exacerbate other known issues or introduce new ones.

Currently, the authors support their revision primarily by showing subtle improvements in C–H and H–H contact distance distributions during high-resolution refinement tasks. However, in benchmark tests assessing correlation with experimental ΔΔG measurements, the performance of the revised model is indistinguishable from that of the original. Taken together, these results are insufficient to fully rule out unintended consequences of the parameter changes, especially the balancing between van der Waals (VDW) packing term and other energetic components.

To provide a more comprehensive evaluation, the revised energy function should be applied to the original set of designed mini-proteins to compare its predictions directly against the deep mutational scanning data. Specifically, it should be assessed how well the updated model predicts various classes of rescue mutations, which should include not only favorable large-to-small substitutions (which alleviate overpacking), but also small-to-large mutations. This would help determine whether the refitting disproportionately favors certain mutation types or leads to overprediction of destabilizing large-to-small changes as beneficial.

Reviewer #2: A common goal for the field of protein design is to learn from failed and successful design campaigns so that future attempts have higher success rates. This is an interesting study that uses site saturation mutagenesis of large numbers of de novo designed proteins to learn what went right and wrong during design. The aim was to determine if the energy function used during the design process has systematic pathologies that prevent choosing the most favorable amino acid at each sequence position. Primary findings include the observation that the designed proteins can most often be stabilized by mutating a partially buried polar amino acid to a non-polar amino acid, and that the energy function failed to detect destabilizing clashes at a small set of sequence positions. The second observation was used to guide parameterization of the Rosetta Lennard-Jones potential. This paper will be of general interest to the protein design field. Minor revisions that address the following questions are recommended.

1) The differences made to the LJ potential (table S2) appear to be very small. It would be helpful to include some figures that plot the new potential over the previous potential (with all weights applied)

2) One concern when increasing repulsive forces is that it will lead to underpacking at some residue positions during design. What evidence is there that the changes implemented here do not lead to underpacking?

Reviewer #3: This manuscript presents a novel and systematic approach for improving biomolecular energy functions by learning from the failures of de novo protein designs. The following minor issues should be addressed to enhance clarity:

1. Lines 150–151 state: “We also performed biological replicates of the entire experiment for 19 of the 21 outlier designs.” Could the authors clarify why biological replicates were carried out for only 19 of the 21 outlier designs?

2. Lines 343–344 state: “Following optimization with dualoptE, we manually fine-tuned parameters to further improve atom-pair distribution fits.” Could the authors briefly explain the procedure for manual parameter fine-tuning?

3. The study focuses primarily on retraining the beta_jan25 energy function. Would it be more appropriate to present data for beta_jan25 rather than beta_nov16 in Figure 5A?

4. Some steric-clashing issues may not be fully resolved by the dualoptE protocol and the Lennard-Jones potential. Could alternative functional forms or protocols be considered? The authors may wish to briefly discuss this possibility.

**Have the authors made all data and (if applicable) computational code underlying the findings in their manuscript fully available?**

The PLOS Data policy requires authors to make all data and code underlying the findings described in their manuscript fully available without restriction, with rare exception (please refer to the Data Availability Statement in the manuscript PDF file). The data and code should be provided as part of the manuscript or its supporting information, or deposited to a public repository. For example, in addition to summary statistics, the data points behind means, medians and variance measures should be available. If there are restrictions on publicly sharing data or code —e.g. participant privacy or use of data from a third party—those must be specified.requires authors to make all data and code underlying the findings described in their manuscript fully available without restriction, with rare exception (please refer to the Data Availability Statement in the manuscript PDF file). The data and code should be provided as part of the manuscript or its supporting information, or deposited to a public repository. For example, in addition to summary statistics, the data points behind means, medians and variance measures should be available. If there are restrictions on publicly sharing data or code —e.g. participant privacy or use of data from a third party—those must be specified.requires authors to make all data and code underlying the findings described in their manuscript fully available without restriction, with rare exception (please refer to the Data Availability Statement in the manuscript PDF file). The data and code should be provided as part of the manuscript or its supporting information, or deposited to a public repository. For example, in addition to summary statistics, the data points behind means, medians and variance measures should be available. If there are restrictions on publicly sharing data or code —e.g. participant privacy or use of data from a third party—those must be specified.requires authors to make all data and code underlying the findings described in their manuscript fully available without restriction, with rare exception (please refer to the Data Availability Statement in the manuscript PDF file). The data and code should be provided as part of the manuscript or its supporting information, or deposited to a public repository. For example, in addition to summary statistics, the data points behind means, medians and variance measures should be available. If there are restrictions on publicly sharing data or code —e.g. participant privacy or use of data from a third party—those must be specified.

Reviewer #1: Yes

Reviewer #2: Yes

Reviewer #3: None

PLOS authors have the option to publish the peer review history of their article (what does this mean?). If published, this will include your full peer review and any attached files.). If published, this will include your full peer review and any attached files.). If published, this will include your full peer review and any attached files.). If published, this will include your full peer review and any attached files.

...

Reviewer #1: No

Reviewer #2: No

Reviewer #3: No

**Figure resubmission:**
---

## [Decision Letter · Decision Letter 1]

9 Apr 2026

Dear Dr Haddox,

We are pleased to inform you that your manuscript 'Using experimental results of protein design to guide biomolecular energy-function development' has been provisionally accepted for publication in PLOS Computational Biology.

Best regards,

Shi-Jie Chen

Academic Editor

PLOS Computational Biology

Arne Elofsson

Section Editor

PLOS Computational Biology

Reviewer's Responses to Questions

**Comments to the Authors:**

Reviewer #1: In their response letter, the authors report evaluation results of the modified energy function on rescue mutations, which suggest that the modifications do not improve prediction for such mutations. Nevertheless, they have chosen not to include these results in the revised manuscript, arguing that the analyses did not account for potential structural effects induced by the mutations. While this reasoning is plausible, it warrants explicit discussion in the main text, at least as a caveat to the paper’s central claim that “failure modes” observed in de novo protein design can serve as “guiding principles” for retraining energy functions.

Reviewer #3: None.

**Have the authors made all data and (if applicable) computational code underlying the findings in their manuscript fully available?**

The PLOS Data policy requires authors to make all data and code underlying the findings described in their manuscript fully available without restriction, with rare exception (please refer to the Data Availability Statement in the manuscript PDF file). The data and code should be provided as part of the manuscript or its supporting information, or deposited to a public repository. For example, in addition to summary statistics, the data points behind means, medians and variance measures should be available. If there are restrictions on publicly sharing data or code —e.g. participant privacy or use of data from a third party—those must be specified.requires authors to make all data and code underlying the findings described in their manuscript fully available without restriction, with rare exception (please refer to the Data Availability Statement in the manuscript PDF file). The data and code should be provided as part of the manuscript or its supporting information, or deposited to a public repository. For example, in addition to summary statistics, the data points behind means, medians and variance measures should be available. If there are restrictions on publicly sharing data or code —e.g. participant privacy or use of data from a third party—those must be specified.requires authors to make all data and code underlying the findings described in their manuscript fully available without restriction, with rare exception (please refer to the Data Availability Statement in the manuscript PDF file). The data and code should be provided as part of the manuscript or its supporting information, or deposited to a public repository. For example, in addition to summary statistics, the data points behind means, medians and variance measures should be available. If there are restrictions on publicly sharing data or code —e.g. participant privacy or use of data from a third party—those must be specified.requires authors to make all data and code underlying the findings described in their manuscript fully available without restriction, with rare exception (please refer to the Data Availability Statement in the manuscript PDF file). The data and code should be provided as part of the manuscript or its supporting information, or deposited to a public repository. For example, in addition to summary statistics, the data points behind means, medians and variance measures should be available. If there are restrictions on publicly sharing data or code —e.g. participant privacy or use of data from a third party—those must be specified.

Reviewer #1: Yes

Reviewer #3: None

PLOS authors have the option to publish the peer review history of their article (what does this mean?). If published, this will include your full peer review and any attached files.). If published, this will include your full peer review and any attached files.). If published, this will include your full peer review and any attached files.). If published, this will include your full peer review and any attached files.

...

Reviewer #1: No

Reviewer #3: No

---

## [Editor Report · Acceptance letter]

PCOMPBIOL-D-25-02655R1

Using experimental results of protein design to guide biomolecular energy-function development

Dear Dr Haddox,

I am pleased to inform you that your manuscript has been formally accepted for publication in PLOS Computational Biology. Your manuscript is now with our production department and you will be notified of the publication date in due course.

With kind regards,

Aiswarya Satheesan
